# Prevalence of Pain within Limb Girdle Muscular Dystrophy R9 and Implications for Other Degenerative Diseases

**DOI:** 10.3390/jcm10235517

**Published:** 2021-11-25

**Authors:** Mark Richardson, Anna Mayhew, Robert Muni-Lofra, Lindsay B. Murphy, Volker Straub

**Affiliations:** John Walton Muscular Dystrophy Research Centre, Newcastle University and Newcastle Hospitals NHS Foundation Trust, Newcastle upon Tyne NE1 3BZ, UK; mark.richardson25@nhs.net (M.R.); anna.mayhew1@nhs.net (A.M.); robert.muni@nhs.net (R.M.-L.); lindsay.murphy@newcastle.ac.uk (L.B.M.)

**Keywords:** pain, neuromuscular diseases, LGMDR9, limb girdle muscular dystrophy, pain assessment, pain management

## Abstract

Our primary aim was to establish the prevalence of pain within limb girdle muscular dystrophy R9 (LGMDR9). As part of the Global FKRP Registry, patients are asked to complete the Short Form McGill Pain Questionnaire (SF-MPQ) annually. We used the results of this questionnaire to determine individuals’ maximum pain score and total pain score and examined overall pain intensity and associations between pain intensity and LGMDR9 genotypes, age, and ambulatory status. We also considered the pain descriptors used and pain progression over time. Of the 502 patients, 87% reported current pain and 25% reported severe current pain. We found no associations in pain severity between the different genotypes of LGMDR9. However, we did find statistically significant associations between pain severity and ambulatory status and between our paediatric and adult populations. We found pain descriptors to be more common words that one may associate with non-neural pain, and we found that a significant number of individuals (69%) reported a fluctuating pain pattern over time. We concluded that pain should be considered a significant issue among individuals with LGMDR9 requiring management. Implications regarding assessment of pain for other degenerative diseases are discussed.

## 1. Introduction

Muscular dystrophies are a group of genetic neuromuscular diseases (NMD) characterised by progressive muscle weakness and wasting and dystrophic changes in skeletal muscle tissue, with loss of normal muscle fibres and their replacement by fat and connective tissue. These diseases can often have devastating effects on a person’s quality of life and in many cases on life expectancy [1].

Traditionally, pain has not been regarded as a prominent symptom in most genetic NMD, overshadowed by progressive muscle weakness and its impact on function, but in recent years, a growing number of studies have begun to investigate the prevalence and impact of pain within NMD [2,3,4,5,6,7,8,9,10,11,12,13,14,15,16,17,18,19,20,21,22,23].

It is widely regarded that facioscapulohumeral muscular dystrophy (FSHD) is one of the more painful conditions within the group of muscular dystrophies. Morís and colleagues reported that of 383 patients with FSHD, 339 (88.5%) reported experiencing current pain [19]. Other studies looking into current pain within FSHD found similar percentages, ranging from 76% to 89% [20,21,22]. With regard to other NMD, in a sample of 130 patients with myotonic dystrophy, 78 (60%) reported pain [20]. Of subjects with pain, 19 (24%) reported their pain as severe. Similarly, Guy-Coithard and colleagues reported on 128 patients with Duchenne or Becker muscular dystrophy and found that 85 (66%) experienced pain, 19% of whom described their pain as severe [21]. In a large survey (*n* = 617) involving Charcot-Marie-Tooth patients, 440 (71%) reported having pain, 171 (39%) of whom described the pain as severe [2]. 

Thus far, there appear to be no published data considering the prevalence of pain within different types of limb girdle muscular dystrophy (LGMD) [24], although Jensen and colleagues did report that of 44 patients with LGMD, 64% reported current pain, 25% of whom classified their pain as severe [22]. However, this study did not differentiate between types of LGMD.

Comparisons between these data are difficult due to differences in the recall period used and methods of data collection; however, it appears that pain is a significant problem for a substantial subset of individuals with NMD.

In terms of how these figures compare to the general population, a recent systematic review looking specifically into chronic pain found that the national UK prevalence of chronic pain is 43%, with an estimated prevalence of severe pain ranging between 10.4% and 14.3% [25]. These figures would suggest that patients with NMD do experience a greater prevalence of pain than the general population. The mechanism behind pain within NMD remains largely unexplored, although various theories have been suggested, especially for inherited neuropathies, where pain may be either mechanical or neuropathic [2].

LGMD, for which a revised classification system has recently been developed [26], includes one subtype, LGMDR9 (formally 2I), which is caused by mutations in the Fukutin-Related Protein gene (*FKRP*) [27], in which individuals are most commonly homozygous for a Scandinavian founder mutation [28]. Less frequently, individuals may be compound heterozygous for the founder mutation or may be homozygous or compound heterozygous for other mutations [29]. This information is important because patients who are homozygous for the founder mutation tend to have a milder phenotype and later disease onset when compared to patients who are compound heterozygous for the common mutation or those who have a mutation other than the common mutation [29]. 

Individuals with LGMDR9 show both muscular dystrophy and in many cases can develop cardiomyopathy and respiratory problems. Natural history data for this condition have primarily focussed on muscle weakness and function and muscle pathology [30,31] but as of yet has not considered pain as a prominent component of the disease.

There is limited information on the severity or type of pain within patients with LGMDR9 and how this compares to the normal population or with other NMD. Therefore, the primary aim of this study was to establish the prevalence of pain within LGMDR9 and to explore associations between severity of pain and LGMDR9 type, age, and ambulatory status. The second aim was to establish if common pain descriptors that LGMDR9 patients attach to their pain provide any clues as to the underlying mechanisms for pain. The final aim was to explore if pain experienced by those with LGMDR9 exhibits a particular pattern of progression.

## 2. Materials and Methods

Data were obtained from the Global FKRP Registry. This is an international, online registry, accessed at www.fkrp-registry.org (accessed on 28 June 2021), which captures clinical and genetic information entered by both patients and their nominated clinicians via the same interface. Patients initiate registration and provide consent online. They are then asked to complete a short mandatory questionnaire about the current status of their health and two optional validated questionnaires on quality of life and pain (the short form of the McGill pain questionnaire; SF-MPQ) (copyright for questionnaire use was obtained). They are requested to complete this health questionnaire on an annual basis.

The SF-MPQ is a widely used, well validated, and reliable measure of pain and is frequently the measure of choice when considering pain within NMD [19,21,22]. The first question of the SF-MPQ consists of 15 pain descriptors (11 sensory and 4 affective), which are rated on an intensity scale ranging from ‘none’ through to ‘mild’, ‘moderate’, and ‘severe’. The user is asked to circle which intensity response is the most appropriate for each pain descriptor. We focussed our analyses on this first question of the SF-MPQ only. The recall period for reports of pain was within the last one week.

Here, we provide cross-sectional and longitudinal data analysis on individuals with both genetically confirmed and unconfirmed LGMDR9.

### Analysis

Using an individual’s most recent questionnaire, we conducted Chi-square analyses to compare pain severity with LGMDR9 genotype, age group, and ambulatory status. Results were considered significant at *p* < 0.05. Age groups were amalgamated to define paediatric (≤17 years) and adult (18+ years) populations. For pain, the individual’s maximum pain was reported, regardless of the descriptor used, scoring no pain as zero, mild pain as 1, moderate pain as 2, and severe pain as 3. Individuals who had not reported their ambulatory status in their most recent questionnaire were excluded.

The percentage of individuals reporting each pain descriptor, regardless of severity, was calculated using the individual’s most recently completed questionnaire.

For longitudinal analysis, patients who filled in only one or two questionnaires were excluded. To calculate an overall pain score, we used the sum total score of question 1 of the SF-MPQ (0–45). Change in this pain score was examined between questionnaires to establish if an increasing, decreasing, fluctuating, or no change pain trend existed.

## 3. Results

### 3.1. Demographic Information

Five hundred and two patients with LGMDR9 were reviewed from data collected from the registry’s inception in March 2011 through to May 2021. Of these patients, genetic confirmation was pending for 267 patients. The patients originated from 41 countries. Demographic information regarding LGMDR9 genotype, age, ambulatory status, and sex is presented in Table 1.

### 3.2. Overall Pain Severity

Eighty-seven percent of LGMDR9 patients reported experiencing pain within the last one week of filling in their most recent questionnaire. Twenty-five percent of the patients reported this pain to be severe (Figure 1).

### 3.3. Associations between Pain Severity Categories and LGMDR9 Type

For individuals with a confirmed LGMDR9 diagnosis, those with a milder phenotype (homozygous for the common mutation) had a greater prevalence of pain (88%) than those with more severe phenotypes (not homozygous for the common mutation) (79%) (Figure 1). However, the association between LGMDR9 genotype and severity of pain was not found to be statistically significant for this group ($\chi_{3}^{2}=3.56,\;$ *p* = 0.314). In addition, the ‘homozygous’ and the ‘not homozygous’ groups exhibited the same percentage of severe pain (21%) (Figure 1).

Individuals with an unconfirmed diagnosis reported similar percentages of pain to those with a confirmed diagnosis (either homozygous or not homozygous for the common variant) (Figure 1). The association found between pain severity and whether the diagnosis was confirmed or unconfirmed was not found to be statistically significant ($\chi_{3}^{2}=7.34,\;$ *p* = 0.062). As no significant differences in pain incidence and severity were observed between patients with confirmed and unconfirmed LGMDR9, subsequent analyses presented in this paper were performed on the patient group as a whole.

### 3.4. Associations between Pain Severity Categories and Age

The association between pain severity and the various age categories was not found to be statistically significant ($\chi_{21}^{2}=24.77,\;$ *p* = 0.2575). However, additional analyses merging the paediatric population and adult population did find a significant association ($\chi_{3}^{2}=9.11,\;$ *p* = 0.028). Paediatric patients were more likely to have more reports of ‘no pain’ and fewer reports of ‘severe pain’ than adult patients (Figure 2).

### 3.5. Associations between Pain Severity Categories and Ambulatory Status

Individuals who reported their ambulatory status as being able to walk (unsupported) reported the greatest prevalence of overall pain (97%) and also the greatest prevalence of severe pain (32%). Pain severity and ambulatory status had a statistically significant association ($\chi_{18}^{2}=33.5,\;$ *p* = 0.015) (Figure 3). Those who were able to walk unsupported were more likely to report overall and severe pain than those in the other categories. In addition, individuals with the greatest level of mobility (able to run) were more likely to have no pain than individuals with less motor function (Figure 3).

### 3.6. Pain Descriptors

Figure 4 illustrates the breakdown of pain descriptors. Tiring and aching pain were the most commonly reported type of pain.

### 3.7. Pain Progression

In all, 138 completed three or more questionnaires, and 100 of them had genetic confirmation of a diagnosis. Of these, 69% had a fluctuating progression of pain, 20% of patients were found to have an increasing trend, 6% a decreasing trend, and 5% no change.

## 4. Discussion

Our primary aim was to establish the prevalence of pain within LGMDR9 and to explore associations between the severity of pain and LGMDR9 genotype, age, and ambulatory status. We also sought to establish if there are common pain descriptors and patterns of pain progression over time.

Our study demonstrated that pain is highly prevalent in LGMDR9. When compared to the presence of chronic pain within the general population, the figures are roughly double for both overall pain and severe pain. With respect to specific comparisons within the field of NMD, we found our data to be most consistent with FSHD. Studies examining pain within FSHD had a prevalence range of 76–89% [19,20,21,22]. Our finding of 87% overall pain prevalence fits within these FSHD pain ranges. We conclude that LGMDR9 should indeed be regarded as a painful condition on a par with FSHD.

We found no associations between pain severity and confirmed or unconfirmed cases. Surprisingly, individuals with the milder phenotype (homozygous) experienced more pain than those with the more severe phenotype (not homozygous). This may be affected by the disease stage at which patients were measured.

Individuals who had a ‘mid-level’ mobility status of being able to walk unsupported (but unable to run) reported the greatest amount of overall and severe pain. These associations were found to be statistically significant. Individuals with both greater mobility (e.g., able to run) and lower levels of mobility (e.g., non-ambulant individuals) were less likely to report overall pain and severe pain than individuals who can walk unsupported. The implications of these findings to those with the disorder are unclear. Ambulant individuals who walk without aids may be keen to keep walking as long as possible, but perhaps walking aids or spending less time on their feet may have a positive impact on their pain.

Adult individuals with LGMDR9 are more likely to experience overall and severe pain than the paediatric population. Nevertheless, it would be inaccurate to conclude that pain is not an issue for children. Pain management services should be available to all LGMDR9 individuals who require it regardless of their age.

The most common words used to describe LGMDR9 pain were ‘tiring’ and ‘aching’ pain. Words more akin with neuropathic-type pain were less commonly described. This may provide some suggestion as to a mechanism of pain, but more work would be recommended before any firm conclusions are drawn. Existing pain measurement tools often neglect to capture additional components of patients’ pain. Information concerning the behaviour of pain, mechanism of onset, overall duration, and pain progression patterns, for example, are often lacking and may be helpful to further describe and ultimately manage individuals.

The finding of a high percentage of fluctuating pain could be considered somewhat of a surprising result in a progressive condition and may have significant implications in terms of pain management. Whether this is unique to LGMDR9 or transferable to other pain-producing conditions is unclear. That the magnitude of these fluctuations was not considered could be deemed a limitation and worthy of consideration for future studies.

Although we may be able to conclude that pain is a significant issue in many types of NMD, there is no clear consensus regarding the best way to manage it. Previous studies have cited different options that different patients have found beneficial [7,19,20,22,23]. It seems that not all treatment modalities benefit all patients and, furthermore, some modalities benefit some yet aggravate others [7]. One explanation for this could be the influence of confounding factors. Our finding of 87% pain prevalence may include individuals whose pain may not be directly related to their muscular dystrophy. An attempt to investigate and exclude confounding factors would be a useful future consideration.

The SF-MPQ did not allow us to gather information regarding the location of patients’ pain. Since LGMDR9 is in the family of limb girdle muscular dystrophies, we may hypothesise that pain may have been local to the proximal limb joints, as opposed to distal. This could certainly be a consideration to include for future studies. A final consideration for future studies could be to investigate how pain influences other aspects of patients’ health, such as quality of life and mental health status.

To summarise, pain would appear to be a significant problem within LGMDR9. The presence of pain does not appear to be associated with LGMDR9 type. Although we did find an association between pain severity and ambulatory status, it would appear that this association is not necessarily related to the greatest or least ambulatory status. We did find that LGMDR9 adult patients are more likely to experience pain than LGMDR9 children. Pain did not appear to present itself as neurogenic and is most likely to fluctuate in intensity over time. We propose a number of applicable learning points to the wider pain community. Most notably, there is a need to discuss and produce new assessment tools that could be used in the clinic and the clinical trial settings for pain management and research purposes, respectively. Following this, there is a pressing need to attempt to establish causation of pain that may lead to improved pain management pathways.

## Figures and Tables

**Figure 1 jcm-10-05517-f001:**
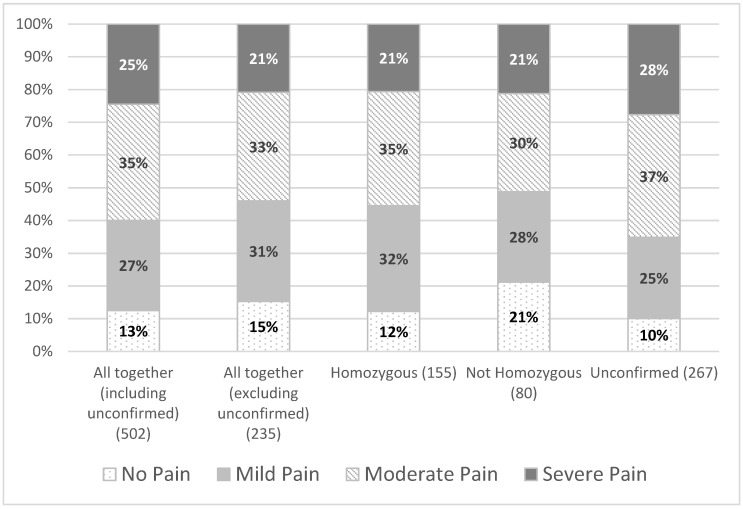
Severity of pain by LGMDR9 type.

**Figure 2 jcm-10-05517-f002:**
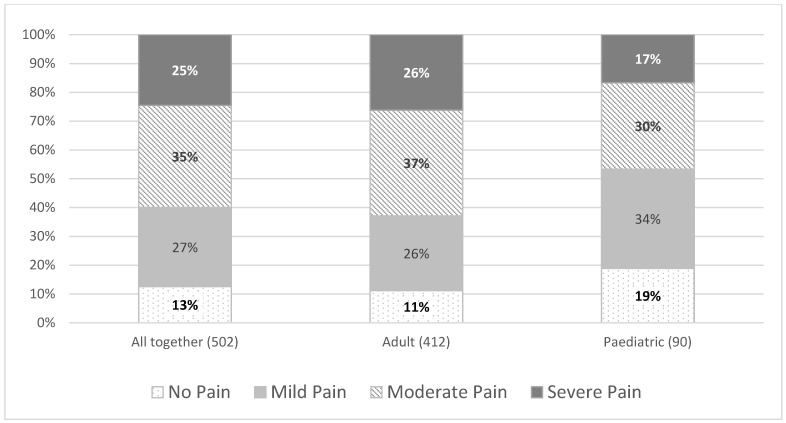
Severity of pain by age; paediatric versus adult.

**Figure 3 jcm-10-05517-f003:**
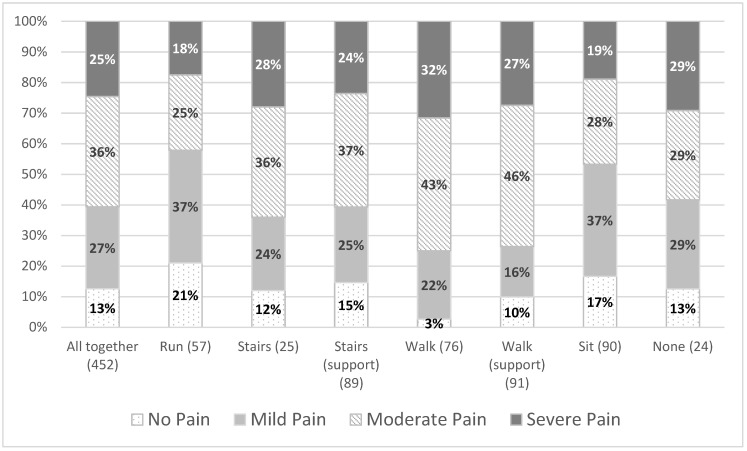
Severity of pain by ambulatory status.

**Figure 4 jcm-10-05517-f004:**
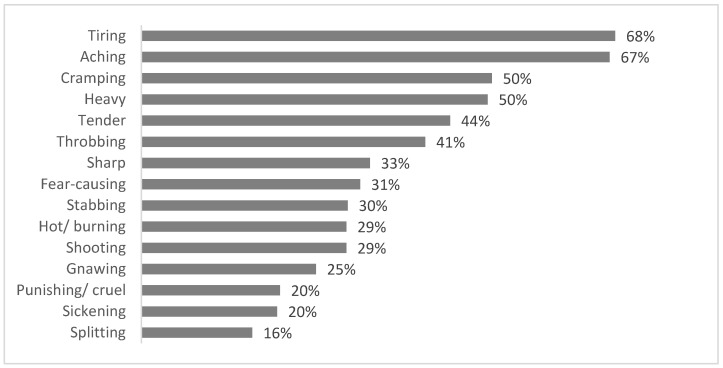
Percentage of individuals reporting each pain descriptor.

**Table 1 jcm-10-05517-t001:** Demographic information.

	Homozygous	Not Homozygous	Unconfirmed	Total
**Sex**				
Female	88	45	154	287
Male	67	35	113	215
**Age**				
0–9	3	9	23	35
10–19	16	21	33	70
20–29	18	16	57	91
30–39	34	7	48	89
40–49	31	11	51	93
50–59	31	11	35	77
60–69	15	4	16	35
70–79	7	1	4	12
**Ambulatory status**				
Run	17	9	31	57
Stairs	10	3	12	25
Stairs (support)	32	10	47	89
Walk	12	13	51	76
Walk (support)	33	7	51	91
Sit	26	21	43	90
Unable to sit independently (none)	5	8	11	24

## Data Availability

Data were gleaned from the Global FKRP Registry.

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
