# Peer review of "Prevalence of Pain within Limb Girdle Muscular Dystrophy R9 and Implications for Other Degenerative Diseases"

_jcm, 2021, doi:10.3390/jcm10235517_

Round 1
Reviewer 1 Report
Authors aimed to establish prevalence of pain within limb girdle muscular dystro
phy R9 (LGMDR9) in 502 patients using Short 11 Form McGill Pain Questionnaire (SF-MPQ) on an annual base. 87% reported current pain and 25% reported severe current pain.Significant association were found between pain severity and ambulatory status and between paediatric and adult
populations.
This study adresses an important issue of pain in the context of a muscular distrophy. Although no clinical examination took place, this questionnary analysis provides new insights into the subject. It would be useful to include a subsection on pain management in these patients, if there are any guidelines on the subject, especially in the pediatric population.
Furthermore, pain localization (whole body pain, joint pain, shoulder/hip pain) and discussion around modalities of conservative treatment should be included. What is lacking to this article are possible confounding elements; i.e. the patients with distrophy may have other conditions which lead to pain. If possible, please include either in your results new data on pain localization or subsection in discussion on pain localization. Are there any data which support increased frequency of herniated discs or spinal canal stenosis, i.e. degenerative spine diseases in the context of muscular distrophy?
Author Response
We are grateful for your helpful feedback and have addressed your comments point-by-point below.
- It would be useful to include a subsection on pain management in these patients, if there are any guidelines on the subject, especially in the pediatric population.
We agree that a subsection on pain management would be useful. To our knowledge, there are unfortunately no published guidelines, but many suggestions of management options in some of the referenced articles. We have now elaborated on this in the discussion section (see lines 237-241)
- pain localization (whole body pain, joint pain, shoulder/hip pain) and discussion around modalities of conservative treatment should be included. Please include either in your results new data on pain localization or subsection in discussion on pain localization.
We agree that pain localisation would indeed greatly add to the study but this unfortunately a limitation of the questionnaire itself. We have added a sentence to this effect in the discussion. (see lines 245-248)
- What is lacking to this article are possible confounding elements; i.e. the patients with dystrophy may have other conditions which lead to pain. Are there any data which support increased frequency of herniated discs or spinal canal stenosis, i.e. degenerative spine diseases in the context of muscular dystrophy?
We absolutely agree that a proportion of our patients reporting pain could have had confounding factors. Specific data on confounding factors have not been captured in the pain questionnaire or the Global FKRP Registry. To this end, we have now added this as a useful direction for future study. (see lines 241-244)
Reviewer 2 Report
This is an interesting study of the prevalence and descriptive characteristics of pain in LGMDR9, and its association with factors like age, ambulatory status, etc. Overall the methods are sound and the conclusions important for practicing neuromuscular clinicians. I would like to know if the authors have any data on a) the impact of pain in overall quality of life in LGMDR9 (assessed by e.g. SF-36 scores) and b) the association of pain intensity/severity with depressive symptomatology (eg. PHQ9 scores)
Author Response
We are grateful for your helpful feedback and have addressed your comment below.
- I would like to know if the authors have any data on a) the impact of pain in overall quality of life in LGMDR9 (assessed by e.g. SF-36 scores) and b) the association of pain intensity/severity with depressive symptomatology (eg. PHQ9 scores)
We agree that data such as this would be a useful addition to the study. With this in mind, we have added a sentence suggesting it as a consideration for future study. (see lines 248-250)